# Soundtracking the Public Space: Outcomes of the Musikiosk Soundscape Intervention

**DOI:** 10.3390/ijerph16101865

**Published:** 2019-05-27

**Authors:** Daniel Steele, Edda Bild, Cynthia Tarlao, Catherine Guastavino

**Affiliations:** 1School of Information Studies and CIRMMT, McGill University, Montreal, QC H3A 0G4, Canada; cynthia.tarlao@mail.mcgill.ca (C.T.); catherine.guastavino@mcgill.ca (C.G.); 2Department of Geography, Planning and International Development, University of Amsterdam, 1018WV Amsterdam, The Netherlands; A.E.Bild@uva.nl

**Keywords:** Musikiosk, soundscape, sound perception, soundscape intervention, democratic soundscape installation, quality of the urban public experience, mixed methods study, pocket park, urban sound planning

## Abstract

Decades of research support the idea that striving for lower sound levels is the cornerstone of protecting urban public health. Growing insight on urban soundscapes, however, highlights a more complex role of sound in public spaces, mediated by context, and the potential of soundscape interventions to contribute to the urban experience. We discuss Musikiosk, an unsupervised installation allowing users to play audio content from their own devices over publicly provided speakers. Deployed in the gazebo of a pocket park in Montreal (Parc du Portugal), in the summer of 2015, its effects over the quality of the public urban experience of park users were researched using a mixed methods approach, combining questionnaires, interviews, behavioral observations, and acoustic monitoring, as well as public outreach activities. An integrated analysis of results revealed positive outcomes both at the individual level (in terms of soundscape evaluations and mood benefits) and at the social level (in terms of increased interaction and lingering behaviors). The park was perceived as more pleasant and convivial for both users and non-users, and the perceived soundscape calmness and appropriateness were not affected. Musikiosk animated an underused section of the park without displacing existing users while promoting increased interaction and sharing, particularly of music. It also led to a strategy for interacting with both residents and city decision-makers on matters related to urban sound.

## 1. Introduction

The deleterious effects of noise on everyday life (particularly in urban settings) is well established. Organizations at national, international, and supranational levels have (rightfully) emphasized the negative effects of noise on public health and have emphasized the “cost of noise pollution” (See e.g. Ireland: http://www.askaboutireland.ie/enfo/irelands-environment/noise/cost-of-noise-pollution/). While the field of noise abatement has achieved dramatic reductions in the presence of sounds in urban spaces that exceed harmful sound levels or cause excessive annoyance, it remains impossible, as well as undesirable, to conceive of a city that makes no sound. Today’s approaches, heavily reliant on the measurement of sound level (and its subsequent reduction) offer little guidance on identifying what sounds belong in the city once the harmful sources have been mitigated. We suppose that sounds are a reflection of public life [1], playing a complex role in our urban experience. In the sound-oriented research community, there is a growing understanding that exclusive reliance on physical measurements, and particularly sound levels, fails to capture the complexity of the human experience (e.g., [2,3,4,5]), providing limited guidance to planners and designers on identifying or designing better sounding cities.

Conversely, an extensive and ever-growing body of research on soundscape (defined as the acoustic environment as perceived, experienced, and/or understood by people or society, in context [6]) offers a new approach for urban sound management, centered on the idea of sound as an urban resource [2,7,8,9,10,11]. Studies embedded in this tradition are based on the experience of the city user and on a complex understanding of the urban auditory experience, repeatedly demonstrating that urban soundscape strongly influences overall health but not always in a negative way (see Aletta et al. [7] for a review of positive outcomes, including stress recovery [12,13]). Despite this knowledge collected in the research community, urban sound management remains driven by noise abatement strategies, on both the city-maker side (i.e., planners, politicians) [14] and the city-user side (i.e., communities, stakeholders) [15].

To investigate the role of sound in urban experience, we deployed a soundscape intervention in the gazebo of a Montreal public park. Musikiosk provided free, unsupervised access to an audio jack and speakers. It allowed people to play whatever audio content they wished in the public space. A stakeholder-based approach as well as rules set forth, described in the Methods section, started in advance of the intervention were intended to minimize risks to the project, as described in the Results section. This study addresses the following research question: “How does a soundscape installation like Musikiosk affect the quality of the urban public experience for users of a public space?”.

To this end, we introduce the hybrid concept of “quality of the urban public experience” (QUPE) as a bridge between public health and the present topic of this special issue, soundscape. The QUPE concept comprises themes that are both operationalizable and measurable, providing the opportunity to empirically research the effects of a democratic soundscape installation on the users of a pocket park in Montreal.

Musikiosk was developed based on ideas of sound as a resource and of providing park users with the opportunity (and responsibility) of sonically appropriating their space. We bring into the conversation insights from a number of social and human domains of science (including sociology, psychology, acoustics, soundscape research, and public space research) to define QUPE. To date, the positive link between the conceptualization of “sound as a resource” [4] and public health has been posed as an obvious one, yet there are no available models to utilize.

QUPE thus serves as a catch-all, multidisciplinary concept into which we can integrate the complexity of the psychological, behavioral, and other social aspects of urban space use, with a focus on the quality, including sound quality, of the interaction between city users and their urban public spaces. QUPE (see Figure 1) is operationalized along three axes (which for the purposes of this paper are most elaborated in the sonic dimension): (1) sound-related evaluation (does Musikiosk change the evaluation of soundscapes and specific sounds?), (2) public space engagement (does Musikiosk impact how users engage in the space and interact with others?), and (3) psychological outcomes (does Musikiosk affect users’ behaviors and feelings?). Within each of these three dimensions, themes related to the presence of Musikiosk emerge and are elaborated in this manuscript.

We situate the study of the aforementioned effects in a specific spatial context i.e., an urban pocket park, with its own physical, social, and acoustic idiosyncrasies, as well as a politico-technological context, referring to the implications of developing and engaging with an installation that encourages a novel, democratic appropriation of public spaces through sound (particularly music).

We report on the results of a mixed-methods study centered on the Musikiosk intervention and highlighting only results that could be elaborated across multiple methods. The QUPE model links the axes of soundscape and public space evaluations to concepts in the public health literature. The Review section steps through this QUPE model, including a necessary background on soundscape evaluations, and other relevant contexts of the study. The Methods section outlines seven different qualitative and quantitative methods, outlining their integration. The Results section elaborates on findings about Musikiosk across multiple methods, organized by the three QUPE axes. The Discussion section situates these results in the context of the QUPE model and other literature while the Conclusions explain theoretical and practical contributions of the study.

## 2. Literature Review

As Musikiosk is primarily an intervention with an effect on the sound of the urban environment, it is important to focus on research that both highlights the ways people evaluate soundscape and soundscape interventions, but also the relationship of soundscape (interventions) to broader aspects of the quality of the public experience, for example, the restorative capabilities of soundscapes. The literature review steps through the axes of the quality of urban public experience (QUPE) outlined in the introduction. This is followed by a review of work on the contexts (i.e., spatial and politico-technological) relevant to Musikiosk.

### 2.1. Sound-Related Evaluation

The first axis of QUPE deals specifically with the evaluation of sound(s) and the sound environment itself.

#### 2.1.1. Soundscape Evaluation

We first review evaluations of the soundscape as a whole, followed by literature on the way people describe specific sounds within the urban sound environment.

##### “Traditional” Soundscape Descriptors

Soundscape descriptors have been proposed to help explain or predict users’ evaluations [16,17] (see [7] for a review). Among these efforts, the Swedish Soundscape-Quality Protocol (SSQP) has stood out as a useful tool to both characterize and compare soundscapes and their perceived affective quality. The scale is composed of a composite of a holistic scale (Good–Bad) [18] paired with a set of eight Likert scales related to *pleasantness*, *unpleasantness*, *eventfulness*, *uneventfulness*, *vibrancy*, *monotony*, *calmness*, and *chaoticness* [19,20]. While originally developed in Swedish, SSQP has been translated in a number of languages, including English, Korean. The in-house French translation, used in this research [21] faced a number of challenges, including the validation of the translation, which led to the *uneventful* rating being dropped.

Recent laboratory and on-site soundscape studies, attempting to further refine soundscape evaluation, have shown the need to add a measurement of the appropriateness of soundscape for a location [7,22,23] and *appropriate* has thus been suggested as a complement to the SSQP ratings [18].

##### Soundscape Appropriateness for Activity

While the role of users’ activities has been suggested almost 20 years ago [3,24] and has also been discussed in relation to the acoustic design of outdoor spaces [25], the integration of activity in models attempting to characterize urban soundscapes is still in its exploratory phase [26,27,28,29,30,31]. Empirical studies offer some evidence on the importance of activities in both laboratory study tasks (e.g., [32,33]) and in-situ soundscape evaluations [34]. However, most such research projects arise from more practice-oriented questions, either dealing with specific soundscape interventions with some form of behavioral control in mind (see [23,26]) or investigating the effect soundscapes play on relaxation or rehabilitation activities or in relation to aspects of indoor and outdoor auditory comfort [35,36,37,38].

In Guastavino’s [32] free sorting tasks of recorded sound environments, participants spontaneously grouped soundscapes in terms of activities, describing them in terms of the actions performed (e.g., “do the groceries”, “take a walk”, “have a drink”) in relation to the type of locations (“market,” “café”, “restaurant,” “park”) and specific sound sources (“vendors,” “music,” “birds”), indicative of the activities. Davies et al. [33], using soundwalks, laboratory studies, and focus groups, found consistently that “soundscapes that are compatible for one’s own purposes and support one’s own behaviors are […] evaluated as positive”. Nielbo et al. [39] further observed that various recorded “soundscapes” were evaluated as being more or less appropriate for different imagined activities by participants in a listening experiment.

Furthermore, recent studies suggest that the level of social interaction of users’ activities also influences how public space users evaluate their soundscapes [31,40]). Therefore, in this paper we understand appropriateness of soundscapes not in relation to a location, but rather to one’s potential or actual activities in the context of outdoor, public space settings [31].

#### 2.1.2. Specific Sounds

In an urban outdoor setting, added sound has been proven to be an efficient masker for other (potentially unpleasant) sounds, and this approach has been used both for purposeful acoustic design, e.g., by installing fountains in parks to reduce annoyance [25,41] or using curated (added) sound art [37,42,43,44]. Added sound has also been used in the context of noise pollution management to, for example, reduce annoyance to mechanical sound [45] or to highway noise [46]. A study on auditory comfort in public spaces showed that introducing sounds that were considered pleasant (such as music and water), even rather loud ones, produced a considerable improvement in evaluations acoustic comfort [47]. Worth noting, is that all of these added sounds were chosen and “composed” by sound designers in a top-down approach. None of the interventions allowed space users to have any control over levels or content. Through the democratic nature of Musikiosk, we are addressing this gap specifically by not predefining what is played, leaving content and even levels (to an extent) up to users.

Music, as another type of added sound, stands out as a sound with an “aestheticizing” effect [48]. Research indicates that music is a factor influencing people’s judgments on their environment, particularly in relation to their soundscape [49,50].

The present study uses the aforementioned modified SSQP, in addition to two appropriateness scales (appropriateness of soundscape for activity and appropriateness of overall sound level for the park), and identification and evaluation of heard sound sources.

### 2.2. Public Space Engagement

The next axis of QUPE explores the way users engage with public space.

Extensive research has been dedicated to exploring the “desirable qualities” of public spaces as well as how to encourage or achieve the well-documented social, health, environmental, and economic benefits that public spaces bring to the users they serve (see [51,52] for comprehensive reviews). However, central to many studies on public spaces are aspects of sociability, interaction, and use that straddle the lines between the physical dimensions of the space (focused on the materiality of space and how aspects of accessibility and usability are encoded in its morphology [53]) and the enactment of publicness in as part of what Yucesoy refers to as the “interactional and experiential space” [54] (p. 2). Research from a variety of human and social sciences have emphasized the deeply social nature of public spaces and much thought is put into their functionality and in the way they are appropriated by users in their everyday activities, with public spaces representing not only the physical but also the social and cultural context for the occurrence of interactions and developing a sense of community [55,56,57,58,59,60,61,62,63].

The ubiquity of digital technologies, personal mobile technologies, on the one hand [64,65,66] and interactive technologies (e.g., [67,68]), on the other, have been consistently transforming the ways in which people engage with their public spaces, how they interact with others and thus blurring the lines between the public and private realm, not only spatially but also acoustically [69,70]. This QUPE axis is operationalized in the present study through observed changes in behavior and interactivity with other users.

### 2.3. Psychological Outcomes

This axis focuses on three psychological outcomes affecting QUPE.

#### 2.3.1. Mood

There is significant support from the literature for a relationship between mood and soundscape. There is a demonstrated negative relationship between sound pressure levels and some aspects of mood (e.g., [71,72]). A relationship between mood (measured in relation to overall annoyance) and evaluations of everyday sounds has been shown in relation to ratings of sound annoyance [73]. For work specifically linking urban soundscape evaluations, Steffens et al. [30,50] demonstrated a significant effect of mood on ratings of pleasantness, eventfulness, and familiarity; this effect was particularly pronounced when music was present during an evaluation.

#### 2.3.2. Public Space Evaluation

The experience of public spaces is moderated at an individual, psychological level by the emotional engagement with the space [74], which in turn affects the future patterns of use, engagement with and evaluation of the space. The interaction with the physical, political, social, or cultural context in general, and with the other users of an urban space in particular can alter one’s personal feelings of safety and security (see [75] for a review), of restoration [76], of leisure [77], of perceived control over the space [78,79], and of psychological comfort (see [57])—often related to one’s perceived freedom of action in a space, of inclusiveness or being excluded from a space ([62], also see [80] for a study on women’s experiences). Mood is measured in the present study with a single, self-reported scale rating.

#### 2.3.3. Soundscape Restorativeness Evaluation

An important aspect of public space experience is the psychological restoration (measured on multiple *restorativeness* scales) that one may achieve from visiting the space [81,82]. Restorative environments enable users to recover from drained cognitive resources and to reflect upon daily- or life issues, and they decrease stress levels [83,84]. Restorative soundscapes, specifically, enable users to recover from the negative effect of noise exposure. The Perceived Restorativeness Soundscape Scale (PRSS) was developed and validated [85,86] to measure the perceived qualities of soundscapes in terms of the four theoretical components considered necessary to create a restorative environment, namely Fascination, Being-Away, Compatibility, and Extent [83,87]. Fascination refers to involuntary, effortless attention. Being-Away involves a shift away from the present situation or problems, allowing tired cognitive structures to rest while activating others. Compatibility refers to the fit of the environment with the needs and inclinations of the user. It has been shown to relate to appropriateness for activity [86]. Extent refers to the richness and structure of the environment into a coherent whole. The first three of these components have been used in the present study due to their relevance in the context of a small urban park.

### 2.4. Spatial Context—Pocket Parks

Ample evidence shows that large urban parks dominated by greenery have measurable effects on aspects of the aforementioned restorativeness as well as stress relief for their users [88,89,90]. However, the question is whether the various health-related benefits apply at smaller scales. The smallest urban public parks, called “pocket parks” [91] by some in the literature, are often as busy as the surrounding city. While these pocket parks have received attention in the design literature, there is not much information available about their potential benefits; they have been left disproportionally understudied, despite intuitive or explicit knowledge on the functions that they serve for the local population [91], including acting as a setting for social interactions and encouraging physical activity in neighborhoods that otherwise lack access to public spaces [92]. A study by Peschardt and Stigsdotter [93] confirms the extensive use of pocket parks for socializing purposes, emphasizing also the effect on mental health and restoration, both for “average” and stressed users. For average users, aspects of socialization were essential, whereas for stressed users, nature was more important, corroborating, to a large extent, the findings for larger green urban parks. A follow up study by the same authors [94] investigated the features of the pocket park that could encourage their potential effect on the two aforementioned health benefits. They showed that different amenities are expected for encouraging socializing and restoration, with some amenities evaluated at times positively for one activity and negatively for the other (like tables or green groundcovers), demonstrating the difficult challenge of designing a small sized public space affording multiple simultaneous uses and purposes. A laboratory-based study [95] showed that, based on visual assessment of pocket parks, they have the potential to afford recovery and restoration-related activities. To our knowledge, the sonic dimension of pocket parks has not been studied; further the function and use of park amenities on the sound environment (and vice versa) is ripe for investigation, though one such study has taken place on a soundscape intervention in a larger public square [96].

### 2.5. Politico-Technological Context

This study also aims to bridge the gap toward understanding user-modified soundscapes. Enabling this study, technological developments of the last decade or two allow listeners to carry their musical choices “in their pocket” and to engage in mobile music listening behaviors in the ephemeral and personalized auditory private spaces delineated by headphones, usually in the context of physical public spaces. These new forms of engagement with urban public through mobile music listening subvert existing norms of social behavior by allowing public space users to control their own auditory experience on site and in real time [97,98]. Given the ample evidence on the effect of music over sociability and loitering behaviors in public spaces (e.g., [26,97,98,99,100,101]), control over music in a public context becomes “a source of social power” [102] (p. 20). Therefore, these behaviors are particularly interesting in light of increased controls set in place in highly monitored public spaces, where rules are continuously created to prohibit a growing list of such behaviors considered disruptive (including playing amplified music in public).

There is a well-documented relationship between music and identity (e.g., [103]), with music as a medium through which people from different cultures and backgrounds can communicate, share experiences and affirm their collective and individual identity. By deciding to bring their private musical behaviors in the public and play their music out loud in a public space setting, users appropriate their space and delineate auditory spaces of identity and belonging—or rejection [100]. In this context, users could engage in ‘responsible listening’ practices [104], complying with shared norms, regulations and behaviors or, on the contrary, defiant to these factors, based on a combination of internal needs (e.g., desire to impress friends or to listen to music) and external considerations (e.g., presence of others). Musikiosk, by encouraging space users to connect their mobile devices to a public output system that is free, accessible, unprogrammed and unsupervised, challenges the centrally set musical policies by promoting new forms of auditory democracy and appropriation. By allowing users to participate in the process of musical production and consumption in public spaces, we revisit the privateness of portable devices, which now become a tool for bringing the private into the public through public islands of music where others can join in.

## 3. Methods

Soundscape research favors qualitative and mixed approaches over quantitative-only approaches [7,105]. The mixed-methods research design developed here follows this trend and was designed to integrate seven different methods at four stages of data collection and a data analysis stage. This integrated approach to data collection and analysis was used to provide a multilayered and holistic understanding of the relationship between park users and their environment as well as the specific effects of Musikiosk on their engagement with the park. All methods were approved by McGill Research Ethics Board, #55-0615.

Figure 2 shows the different methods used at each of the five project phases. The sections below follow the same timeline.

### 3.1. The Musikiosk Device and Parc Du Portugal

Users connected to Musikiosk with anything that had an audio jack (e.g., mp3 players, musical instruments) or a Bluetooth connection (e.g., phone). The device consisted of a Raspberry Pi (Raspberry PI, UK) equipped with a sound card, a Bluetooth module and a Wi-Fi USB antenna (see [106] for a detailed description of the device). It was coupled with an audio card and a calibrated electret microphone for acoustic monitoring. These measurements were used to automatically adjust playback levels to be appropriate based on the time-of-day, automatic system logs for the researchers to monitor usage frequencies, and automatic power down during predetermined nighttime hours (10 PM) as agreed with the city (see [106]).

The Parc du Portugal (see Figure 3) is situated in a lively, musical neighborhood with a history of Portuguese immigration and, more recently, has become one of the centers of Montreal nightlife. On one side of the park is the rue Saint-Laurent commercial artery, home to a large number of different kinds of businesses, active almost 24/7 with shops, cafes, bars, and companies. Yet, on its other side, it borders a quiet(er) residential area, therefore serving a mixed function to both residents and passers-by, like tourists and office workers. Due to the traffic artery and multipurpose activities, there is a rather high background noise level (i.e., 58.5–60.5 dBA during a typical afternoon. The World Health Organization, for example, strongly recommends an average road traffic noise level of below 53 decibels (dB) *L*_den_ (see: http://www.euro.who.int/__data/assets/pdf_file/0009/383922/noise-guidelines-exec-sum-eng.pdf). Because of its location, the park hosts a diverse combination of peacefully overlapping users: recurrent users—elderly members of a Portuguese community who chat with each other from late afternoon through evening, employees from businesses nearby—and also passers-by (see [107] for more details). On the quiet(er) side of this park, a gazebo structure is located, which was chosen for the Musikiosk, because of the history of gazebos as local focal points for music performance, and because of the observed relative lack of use of side of the park. Given this context, the Musikiosk team proposed that a lively musical intervention would not cause a disruption of the park’s character.

The Musikiosk device was unsupervised in that the researchers did not need to be present for it to be operational. Ample signage and instructions were placed around Musikiosk, in addition to a media outreach strategy, including a website and Facebook page, to allow users the opportunity to encounter and use the system. The device was free-of-charge and the democratic aspect of the installation ensured that all users were able to select their own audio content and use any device of their choosing.

### 3.2. Research Design and Timelines

The research project had five distinct phrases spanning more than 18 months: a pre-project, planning phase; Phase A: the pre-intervention; Phase B: intervention; phase C: post-intervention; and a post-project evaluation and analysis phase. The first four of these phases provided the core data reported in this paper. The central phases A, B, and C have been labeled using this separate nomenclature to facilitate comparisons between the experimental conditions arising from the installation of the Musikiosk.

#### 3.2.1. Pre-Project Planning Phase

The project was conceived in the summer of 2014 and the technical aspects were in preparation throughout the following fall and winter. Generating technical and other practical specifications, building the device, selecting and garnering approvals for an installation site, and planning the research took approximately one year. Given Montreal’s weather extremes (very warm summers and very cold winters) and the summer tourist season, the installation was elected to be in place for two months (August and September) in the late summer of 2015.

Given the unusual specifications of the Musikiosk, the permitting process included multiple meetings with municipal employee and elected official stakeholders, including multiple on-site visits, negotiations to liaison with local community and other stakeholders, and technical specifications such as functional hours to be written into the permit. We developed operations rules (e.g., hours of operation, maximal levels) with representatives of the city’s Department of Culture, Sports, Leisure, and Social Development. The permit itself was, in fact, not a permit, but a temporary (2-month) exemption from the noise regulations. Stipulations were appended as conditions of the continued exemption.

During this phase, no formal data were collected; however, observations and accounts from the researchers and details of the permitting process are included to situate parts of the results section.

#### 3.2.2. Phase A—Pre-Intervention

In this phase, formal data collection commenced. A timeframe of approximately one month was chosen for Phase A so that it could be of similar duration to the installation period (Phase B) but would still be in similar conditions for weather and frequency of neighborhood cultural events, enabling controlled comparisons.

In Phase A, three data collection methods were used simultaneously: pre-installation questionnaires (Q1), behavioral mapping and tracking (BMT), and ethnographic observations (EO). After piloting, we conducted 10 days of systematic observations (BMT and EO) in the park to document the patterns of daily park use, the groups of recurrent users and their profiles. Through questionnaires (Q1), we collected a baseline on users’ on-site evaluations of their park and their on-site soundscapes during different times of day and weather conditions. This phase acted as a baseline (control) characterization of the park in terms of use and evaluation (both auditory and overall).

During this period, we also engaged in outreach activities and media engagement, to both advertise the Musikiosk system and the research project (and thus encourage visits and use and engagement of the system), as well as to ensure residents and the local community that their potential concerns on the effect of the system on the neighborhood had been heard and that their potential complaints would be addressed in a timely manner. Particular efforts were made to understand and ensure that existing park users, such as elderly Portuguese-Canadians, were not displaced.

#### 3.2.3. Phase B—Intervention

In Phase B—or the intervention phase—we aimed to collect a detailed characterization of the relationship between park users and their space, with a focus on collecting evidence (if any) on the effect of the Musikiosk system on any potential changes in the aforementioned relationship.

We asked users of the system and non-users (present in the park while the system was in use) to complete questionnaires combining open and closed ended questions on their perception of the park, their evaluation of their soundscape as well as their opinion on the system and its appropriateness for the park (Q2 and Q3). We continued with systematic observations (BMT and EO) of the park to observe whether there were any noticeable changes in the dynamic of the park that could be related to the Musikiosk; these observations were conducted both when the system was in use and when it was not. While initially paper-based, during this phase the observations were conducted by trained data collectors using a handheld device equipped with specially designed software.

Throughout the intervention phase, we continued with our community outreach strategy, organizing special events to increase the visibility of Musikiosk in the neighborhood and demonstrating the usability of the system in an informal setting. The events took place on Tuesday evenings. During the events, through ethnographic observations, we focused on how users engaged with the system and others present, how they negotiated the use of Musikiosk, how other users of the park reacted to the system and what was the overall dynamic of the park during the events.

#### 3.2.4. Phase C—The Post-Implementation Stage

Following the intervention phase, Phase C was an opportunity to speak more in-depth with Musikiosk users (IM) and nearby residents (IR) to reflect on their experience with Musikiosk. These observations were conducted via semi-structured interviews detailed below. In this time frame, Musikiosk was also removed from the park.

#### 3.2.5. Post-Project Evaluation Phase

In this phase, direct feedback to the borough was provided via a written report. Also, feedback on the project outcomes was solicited via interactions with community and researchers. No data from this phase is reported in the present study.

### 3.3. Data Collection

The mixed-methods research protocol developed for this project was designed to be used in an integrated manner. The section that follows details the data collection methods by method and each subsection will describe the procedure, the data collection instrument, and the analysis techniques. Data collection methods have each been given a code (matched in Figure 2) and will be used throughout the document to reference findings specific to the method.

#### 3.3.1. Questionnaires

The purpose of the questionnaires was to collect direct and momentary measurements of participants’ evaluations in use of the park and/or Musikiosk. Questionnaires comprised a combination of open and closed-ended questions (see Appendix A). The questionnaire asked for insight on reported activity (Q1–3), demographic and psychological information (gender, age, noise sensitivity) and other contextual data about users’ visits (e.g., whether they were alone), their evaluation of their soundscape as well as their opinion on the system and its appropriateness for the park.

All 3 questionnaires included 7 of the 8 SSQP scales (pleasant, unpleasant, eventful, vibrant, monotonous, calm, chaotic; Q4). Appropriateness (for activity) was included after the SSQP (Q5). All 3 questionnaires also asked for mood before and during, as self-reported (Q9–10). Mood was probed by directly asking for it (“What is your mood right now?”), and beside the momentary mood, they were asked to make a retrospective judgment (“What was your mood before coming to the park?”). As mood was not the main focus of this study, we feel that using a single mood scale was justified. All of the above scale were 7-point Likert scales.

We collected three types of questionnaires, based on the same template, to mirror the three available park-use conditions. The park baseline during Phase A represents the control condition and is called Q1. During Phase B, with the Musikiosk functioning, questionnaires were given both to those who used the Musikiosk (Q2) and to those who were in the park but did not use Musikiosk (Q3). The questionnaires all measured soundscape evaluation ratings, mood, and demographic data, but they differed only with relation to the details of the condition (degree of Musikiosk engagement). Eighty-eight Q1 questionnaires, 42 Q2 questionnaires, and 67 Q3 questionnaires were administered.

Consistent with the university’s ethics policy, all participants were approached in the park by researchers and asked to take a voluntary, unpaid questionnaire of under 10 minutes. All materials were available in French and English and participants elected to take the questionnaire fully in one language or the other. Participants were approached only after they had already been in the park for a few minutes in order to ensure that they had been exposed the park and its soundscape before making judgments.

##### Q1: Pre-installation Questionnaire (Phase A)

This questionnaire, described above, was administered in the phase preceding the installation of Musikiosk and serves as a baseline condition for the other two conditions. See Appendix A, Table A1.

##### Q2: Musikiosk User Questionnaire (Phase B)

Musikiosk users were given a questionnaire about their experience using it, including who they had chosen to try it, whether it was appropriate for the park, what they liked about the system, and what they thought could be improved. Musikiosk users were not prompted to list the sound sources they heard around them, as were the participants of Q1 and Q3. Participants in this condition were recruited on the basis of having used the system themselves and interacted with their own device to control played content. See Appendix A, Table A2.

##### Q3: Non-MK User Questionnaire (Phase B)

This questionnaire was administered to park visitors who did not physically interact with Musikiosk. This included those who were friends of Musikiosk users as well as those seated on the other side of the park who were unaware of the installation, and all contexts in between. These questionnaires were also administered while the Musikiosk was not necessarily in use.

Participants in this condition were asked what they thought of an installation like this one in this park, how it changed their park experience and activities, whether they found it appropriate, whether it was in use, and whether they would like to use it themselves. See Appendix A, Table A3.

##### Analysis

ANOVA, MANOVA, *t*-tests, and chi-squared tests were performed on quantitative and/or categorized questionnaire data and were dependent on the data type. Relevant statistical methods, results, and significance levels are reported in line in the results text. Qualitative data are counted, categorized into emerging themes using the constant comparison method [108], quoted with descriptions of the analysis in-line and counted. Statistical tests are also reported in detail in Steele et al. [109].

#### 3.3.2. EO: Ethnographic Observations

Ethnographic observations focused on documenting how the existence and use of Musikiosk affected people’s use of their park, with a focus on the changes (if any) of patterns of use of the park by regular users and the patterns of new users. The ethnographic observations were completed both in Phase A (pre-intervention) and Phase B (intervention), concomitantly with the deployment of the other main data collection methods. To this end, we took field notes and photographs and engaged in informal discussions with regular park users.

#### 3.3.3. BMT: Behavioral Mapping and Tracking

To observe the interactions between users of public spaces and their spaces in a non-intrusive yet systematic manner and document their patterns of use of the park, their engagement with amenities, their performed activities on an everyday basis, as well as the groups of recurrent users and their profiles, we used behavioral mapping as a data collection method. Behavioral mapping is the main type of structured observation used to systematically describe, on a map of the space, how its users use and move in it [31,110,111,112]. It allows researchers to integrate spatial and temporal dimensions of use of space and to visually represent changes in patterns of activities. Behavioral mapping was completed throughout the project duration, in each of its three lettered phases (A, B, and C).

For a detailed explanation of the behavioral mapping strategy, please consult [107]. The results of the behavioral mapping were used to offer a behavioral and spatial-temporal context in which we situate the other types of data that were collected, analyzed, and integrated, supplementing the results of the ethnographic observations.

#### 3.3.4. Interviews

Interviews were conducted last, in Phase C (post-implementation), so that an interview guide could be developed in order to include, engage with, and understand emergent findings from the other methods. Residents are labeled R1 through R5 and Musikiosk users U1–U11.

##### IR: Interviews with Nearby Residents

A few weeks to months after the system was taken down, we completed interviews with residents living in the vicinity of the park in which Musikiosk was installed. This included five French-language interviews; two participants (R1 and R2) conducted their interview simultaneously. One of the interviewed residents used Musikiosk themselves and in their interview, we also addressed questions related to their experience of the system. Participants were selected on the basis of living within one block of the park. Their contact information had been collected primarily during outreach activities.

In the interview with residents, we addressed the perceived appropriateness of the Musikiosk system for the park and for the neighborhood as well as how the existence of such a system could change their use of the park. Emphasis was put on whether they observed or experienced any consequences in the use of the park (negative or positive). Questions were also posed as to the greater relationship of these residents with their soundscape for insight on how Musikiosk may have played a role in changing these perceptions.

For the complete interview guide instrument, consult the Appendix B, Table A4.

##### IM: Interviews to Expand on the Questionnaire with Musikiosk Users

Building on the data collected during the questionnaires and ethnographic observations we developed an interview guide aimed to go in depth on how the Musikiosk experience changed the Musikiosk users’ perception and potential future use of the park (if at all) and whether Musikiosk was an appropriate addition for Parc du Portugal. The interview guide included questions in five thematic areas: use of Parc du Portugal, use of the Musikiosk system (including hypothetical scenarios), feedback about the Musikiosk system, a description of the park’s soundscape, and demographic and other background information. Interviews were conducted with 11 users of the system (in English) who had engaged with it throughout the summer. These interviews were intended to be reflective of past uses of Musikiosk and were conducted outdoors in Parc du Portugal.

For the complete interview guide instrument, consult the Appendix C, Table A5.

##### Analysis

The interviews were transcribed and coded by bilingual researchers. They were analyzed using both a combination of deductive coding, based on the operationalization of QUPE, and inductive coding allowing for new themes to emerge. A preliminary list of the themes was originally detailed in [113] but has been updated to reflect the findings matched across methods.

### 3.4. Integrating Methods for Analysis

During the course of the study, some methods were used to inform the design of others. Questionnaires from the first part of Phase A (Q1) informed the variables collected in the behavioral mapping and tracking (BMT) and, to some extent, the recorded ethnographic observations (EO). Results from Q1, Q2, Q3, BMT, and EO informed the Musikiosk user interviews (IM), and each of these subsequently informed the resident interviews (IR).

We first processed and analyzed the data collected in our mixed-methods research design separately and then integrated the key findings along the main dimensions of QUPE (see Table 1), supplemented by additional themes emergent from the different data sources. Methods for each QUPE theme and sub-theme have been labeled as primary when results directly contribute to the theme at-hand and secondary when the results clarify or were informed by the primary methods.

In the presented analysis, themes are presented in sequence, organized roughly by the three QUPE dimensions (sound-related evaluation, public space engagement and psychological outcomes) and respective sub-dimensions. and corroborating findings from each method support the theme at-hand. These themes are detailed at the beginning of the results and form the structure of that section.

## 4. Results


*Results Structure*


We structure our results along the key psychological-behavioral themes of QUPE to illustrate the effects of Musikiosk on the experiences of both Musikiosk and other park users. The results and subsequent discussion are based on integrated analysis of the different methods. We group the aforementioned QUPE themes along three axes: sound-related evaluation; public space engagement; and psychological outcomes. We force, at times, a separation between certain themes for ease of presentation of results. Two additional results sub-sections follow concerning evaluations specifically of Musikiosk (rather than its effects on users) and the lasting effects of Musikiosk after it was uninstalled.

Note: Findings from individual methods indicate the method in parenthesis according to the codes described in Figure 2, e.g., *this finding comes from the pre-installation questionnaire (Q1)*. As some findings have already been published, they are summarized here in the results with references. Author-provided translations of French-language quotations are given in-line with the original in in Appendix D, Table A6, liked using the format *see Table A6–1* to direct the reader to quotation 1. Quotations are reported as, “*sample*” (participant ID, reported gender, age).

### 4.1. Park Profile

An initial integration of results (BMT, EO, Q1, Q2, and Q3) allowed for a characterization of the park in terms of its patterns of use and users, confirming initial observations that it was both a transition and a destination space, the latter particularly for neighborhood users.

BMT and EO pointed to the idea that there were strong differences in park use between the daytime and evening, as well as the weekday and the weekend (see [107] for a detailed analysis). The differences were visible in terms of profile of users, with younger users visiting the park more in the evening and a mix of regulars (elderly Portuguese-Canadians and returning users on their lunch break) during the day, but also for types of activities performed, with, perhaps unsurprisingly, fewer solitary users in the park in the evenings. In interviews (IM), participants claimed that it was quieter at night and they could hear more people.

Activities involving interacting with others dominate the park throughout times of day and the week, earning Parc du Portugal the label of a “social park” [109,114], however, the nature of the lingering, socially interactive activities changed from daytime to evening due to the profile and age of the users engaged in those activities.

The deeply social nature of the park has been emphasized by both residents and Musikiosk users in interviews; one resident (IR) stated of Parc du Portugal, “it’s a park that seems to be used mostly for community meetings” (R5, M, 34, see Table A6–1), whereas a Musikiosk user (IM) emphasized that “it’s a good place to socialize and meet up with friends, maybe stop for lunch, smoke” (U7, F, 18). Another user argued: “it just seems like a neighborhood park. People just come here, I don’t know, maybe in-between places, but I don’t feel like they come here just […] for the park itself” (U10, F, 27). This neighborhood-like quality, reinforced by the aforementioned presence of regular users, has been evaluated as unwelcoming for new users: “the fact that [the park is] very territorial and people seem to have their little clique and their eyes on the entrance, does make it feel less welcoming” (U2, F, 20s).

Some Musikiosk users debated the friendliness of the park, with one interviewee arguing that it was “a cold park […] It’s a bit like a closed park, it’s not super inviting” (U9, M, 26), while two others focused on its appeal or its potential for activities: “I never thought it was a friendly park because it doesn’t have that much […] entertaining activities to do, I guess. It just it doesn’t look very appealing.” (U1, F, 27); “I just pass it on my way to some other places […] I don’t have any particular reason to just stop here” (U8, M, 21).

Observations also determined that specifically the gazebo of the park was, previous to the installation of Musikiosk, used for very little (BMT). The findings set the stage for a soundscape intervention that, rather than disturbing the quiet, appears to take place in a space of negotiation.

### 4.2. Effects of Musikiosk on QUPE

The following sections step through the effects that Musikiosk had on each of the three QUPE dimensions and present emergent themes as they are supported from findings in various methods.

#### 4.2.1. Sound-Related Evaluation

The first of these effects on QUPE pertains to evaluations the soundscape as well as individual sound sources.

##### 4.2.1.1. Soundscape Evaluation

The section on soundscape evaluation has been divided into traditional soundscape descriptors and soundscape appropriateness for activity.


**“Traditional” Descriptors**


The first theme is centered specifically on evaluations of the soundscape using semantic scales. These essentially include 7 of the 8 SSQP descriptors (not including uneventful): pleasant, unpleasant, eventful, calm, monotonous, vibrant, and chaotic, evaluated using scales. Two additional descriptors, friendly and convivial, are also included.

In Phase B (including Q2 and Q3, users and non-users post intervention), compared to Phase A (Q1 pre-intervention), ratings of the soundscape as *pleasant*, *eventful*, and *vibrant* increased significantly, while *unpleasant* decreased significantly. Looking at Q3 alone, only *pleasantness* increased significantly compared to Q1; all other scales remained unchanged. Users of Musikiosk (Q2) found the park more *pleasant* (M = 6.20 vs. 5.72, *p* < 0.01) and *vibrant* (M = 5.29 vs. 4.48, *p* < 0.01) than non-users (Q3). These findings and associated statistical tests are detailed in [109].

To better evaluate the effect of Musikiosk on the evaluations, we divided non-users (Q3) into two categories, based on the self-reported characteristic of Musikiosk being in-use (ON) and not in-use (OFF). For, Musikiosk ON, the soundscape was evaluated significantly more *vibrant* and less *unpleasant*, than for Musikiosk OFF (vibrant: M(MSKon) = 4.92, M(MSKoff) 3.86, F = 6.46, *p* = 0.014) (unpleasant: M(MSKon) = 1.64, M(MSKoff) = 2.41, F = 4.85, *p* = 0.032). There was no significant difference between Musikiosk OFF and the pre-installation condition (Q1). Overall, the system had no significant effect on ratings of the soundscape as *monotonous*, *calm*, and *chaotic*.

Corroborating the scale ratings, in free-response sections of the questionnaires (Q2 and Q3), users claimed to like the *friendliness* and *conviviality* of the system in addition to its *pleasantness*. Five separate users (Q2) spontaneously described the park as being more eventful with the addition of Musikiosk, corroborating the scale ratings from the SSQP questions. In interviews, Musikiosk users (IM) describe a park baseline as either *pleasant, convivial*, or *nice* (e.g., “it’s a really convivial park and it’s really nice” (U3, M, 30)), as well as the experience of using Musikiosk: “having the music and the sounds of the city at the same time is really nice” (P3, M 30). This theme of “city sound as background” will be explored later in Section 4.2.1.2.

In summary, the effects of the Musikiosk on soundscape descriptors was very localized to the times when it was on; when it was not in use, the park returned to “normal”.


**Soundscape Appropriateness for Activity**


Ratings of soundscape appropriateness for participants’ activities did not change significantly as a result of Musikiosk (Q1, Q2, Q3). In other words, overall, the presence of Musikiosk likely did not change the perceived soundscape appropriateness (as directly rated) for activity across all park users. Appropriateness was consistently rated high (Mean = 5.71 of 7, across all conditions), and was the highest rated of all tested descriptors, including pleasantness. With Musikiosk in the park, the high appropriateness rating remained high.

Non-users (Q3) were asked if and why Musikiosk was appropriate for the park (also covered in Section 4.3). Out of 67 total non-users, the majority (55) found Musikiosk appropriate. Of the 12 non-users who did not find it appropriate for the park, only one had actually heard Musikiosk in use: an older man who indicated on his questionnaire that he was trying to read alone in the evening and found the music “a little too party” for his tastes. Interestingly, a resident (IR) who had used Musikiosk stated in their interview that:
“If I was reading, music could indeed bother me at times. Although I didn’t note that people who were reading around us were bothered. I don’t know if it’s loud enough to be disturbing”.(R3, F, 49, see Table A6–2)


This suggests that users may not be good at judging whether people engaged in other activities find a soundscape appropriate. The same resident said, “it’s a very diverse, lively neighborhood. I think it’s placed so as not to disturb, it’s not too close to the houses, so I think it’s very well integrated” (R3, F, 49, see Table A6–3). These findings suggest that Musikiosk was an appropriate installation for a park that was largely appropriate for its activities in terms of soundscape.

Musikiosk was also capable of enabling activity. One user (IM) described that “it’s nice to have something conducive to conversation” (U11, M, 24).

One other facet of appropriateness what that of the sound level. Users almost uniformly rated the park’s sound level as moderate (3 of 5) but considered that level appropriate (3 of 5), rather than too loud or too quiet (Q1, Q3). However, in both of the interview corpuses (IM, IR), participants spontaneously described the park as noisy, especially in relation to the commercial artery, but also from specific types of users (e.g., “it might be a bit noisy from St. Laurent but over here in general it’s a nice place” [U1, F, 27]). This finding paints a picture of the park where people describe it as noisy, but not in a pejorative sense. Musikiosk thus was welcome in this context because it did not violate expectations, for example, of calm that may permeate in other park spaces. A dichotomy of *noisy* but *quiet* and *appropriate* is described by a Musikiosk user (IM):
“I did not use [Musikiosk] the first time because, as I’ve mentioned, people come here together and enjoy the quiet—it seems like they were enjoying the silence and the quietness. Still, even if it’s noisy all around, it’s still an oasis of quietness in the big city”.(U2, F, 20s)


##### 4.2.1.2. Specific Sounds

The presence of Musikiosk had a two-part effect by (1) reducing the prominence of negative sound sources, and (2) being a positive sound source resulting in a soundscape made up of sources that promoted fascination, attention, and memories of the space.

Results from multiple methods show a positive effect of Musikiosk on reducing the identification and prominence of traffic sounds as negative sound sources. Non-users listed fewer negative mechanical sound sources (73 vs. 50) as well as fewer prominent unpleasant sounds (19 vs. 11) than did pre-installation participants (Q1, Q3), indicating a likely mix of energetic and informational sound masking. This evidence is corroborated in interviews (IR, IM):
“Yes, in any case, the Saint Laurent [boulevard] is so noisy that music will certainly be more pleasant than the traffic noise”;(R5, M, 34, see Table A6–4)
“You can definitely hear the cars, the street is there […] you can hear bits of conversations here and there of people. What else do I hear? Birds, the wind a little bit, squirrels. Yeah, but mostly, what I hear the most are the cars if the music is not playing”.(U10, F, 27)


Musikiosk not only reduced the impact of a negative sound source, namely traffic, but also improved the soundscape and the attention paid to it. Five separate users (Q2) described how the system engaged them to listen in the park when they otherwise may have not. Interviewees (IM) elaborated on their new perspective with Musikiosk as a sound source:
“It’s just really nice having the option to play music […] It just adds a nice dimension to the sound environment”.(U5, M, 28)
“For me, it does really enhance the enjoyment of the park, because anywhere I go, I’m like “oh I should listen to music” and if it’s on headphones, then I don’t really mind but now I get cut off of the sounds around me so having the music and the sounds of the city at the same time is really nice.”(U3, M, 30)
“It really feels like—I kind of feel it’s not a—it is a public place, but at the same time, it’s intimate. It feels like an apartment or something, it feels like a party at someone’s place”(U10, F, 27)
“I feel like it turns your situation almost like a movie so whenever you hear music that’s like not in a room but outside, it adds magic to the place I’d say, it feels really good”.(U7, F, 18)


#### 4.2.2. Public Space Engagement

The second QUPE dimension affected by the installation of Musikiosk was in terms of how users behaved in and engaged with the public space.

##### Behavior


**Observed Public Space Use—Lingering Behavior**


Multiple methods showed that the presence of Musikiosk increased lingering behavior in the park. In the free response portion of the questionnaire, 12 users spontaneously reported wanting to stay longer in the park because of the presence of the system (Q2). This was also confirmed by follow-up interviews (IM), where there was a direct relationship between lingering and whether Musikiosk was in use either by themselves or by others. All interviewees (IM) confirmed that Musikiosk and the opportunity to engage with the space sonically, encouraged them to linger in the park, irrespective of who was playing the music.

The lingering effect was robust across personality types, though the details of engagement were different. A self-defined introvert stated that: “I might find myself a little quiet spot nearby and enjoy […] in the periphery […] I’m allergic to interaction” (U2, F, 20s). In contrast, “[If people played something I liked] I’d probably just be inclined to dance or, [at least,] I would be inclined to certainly stay” (U5, M, 28). Looking back at the questionnaire participants, the average self-reported extraversion was no higher for users than for non-users or pre-installation participants, thus Musikiosk didn’t attract extraversion, but extraversion did affect the details of the engagement.

Lingering behaviors are also directly related with the idea of sharing (discussed below), as Musikiosk is repeatedly evaluated as being best enjoyed with company: “it would definitely be a place to hang out with your friends […] I know some of the students who were there, who decided to leave the bar and come back here and hang out and just listen to some music until it closed” (U6, F, 36).

Observations also demonstrated that Musikiosk led to more dynamic and prolonged uses of the gazebo area of Parc du Portugal (EO), particularly during the evenings and when special events had been organized as part of the community outreach strategy.

Most surprisingly, one of the residents (IR) had used Musikiosk and hoped to have it back again the following year because: “It would be nice for it to come back. I think that, in the summer, it’s all right, because we linger more during the summer, especially in this street. Yes, it can be fun to stop, take a break while shopping or between errands” (R3, F, 49, see Table A6–5).


**Observed Public Space Use—Diversification**


While lingering behaviors increased, changing the park demographic, it was necessary to observe whether existing park users were displaced by this new activity. This analysis begins with questionnaire data. To categorize self-reported activity data, a bottom-up approach elaborated six categories of activities: consuming, relaxing, interacting, moving, observing, and music (See Table 2). Prior to the installation of Musikiosk, there were no reported activities related to music. Once Musikiosk was installed, this number jumped to 14 (not including Musikiosk users). Among only the first five of these activities (without music), a chi-squared test revealed no significant difference, *X*^2^ (4, *N* = 251) = 2.0014, *p* = 0.736, between the types of activities conducted before and after the installation of Musikiosk. This finding suggests that the installation of Musikiosk had little effect on the ongoing activities of the park, only superimposing musical activities on top of the existing uses.

These findings were corroborated and explained in more detail through other methods. For Musikiosk users (IM), diversification referred to two distinct aspects: diversification in terms of users as well as (potential) diversification in terms of activities to be performed in the park.

The diversification of users is related to the novel aspect of the system and the new opportunities it affords for park use: “it would attract more people […] if the project is there for long enough and the people know what’s it about, cause surely you know, it takes time for people to learn about it, to know how to use it, to develop patterns” (U10, F, 27). Another added: “it brought people in, definitely […] When there’s people in a park, people tend to want to be in that park too, which is nicer. And people stop by more, show more interest in it.” (U9, M, 26). One other interviewee (IM) observed just this change in park users that Musikiosk brought about:
“normally, it’s an ambiance of older people who are just kind of hanging out, and then there was a lot of 90s pop being played, which definitely changed it from an ‘old-people-sitting-around’ park to a ‘youngish-but-not-too-young-people-dancing-around-to-cheesy-music-from-their-past’ park”.(U4, M, 25)


The diversification can also be understood in terms of how activities performed in the park could change, especially in the context of planned events:
“I see a lot of people that [go to organized events in parks in the city], and especially if all these young mothers are here all the time with their kids, I’m sure it would work really well amongst that group. It’d be nice for like a mix I guess, it’s like a free-for-all and then sometimes organized”.(U6, F, 36)


One interviewee combined these two ideas, saying, “I liked that there were events planned, so then I would know when to show up while other people would be there. I guessed it’s not really something I would use on my own” (U4, M, 25).

While these insights collected from users (IM) may have suggest a shift in the park demographic, the questionnaire results presented above (Q1, Q3) corroborate observations (EO), showing that previously existing, frequent users were not displaced by Musikiosk; instead, Musikiosk added an extra layer of novel activities to the park. As observed, the regular users did not change their patterns of use of the park, performing the same activities during the same time frames and with more or less the same frequency. Furthermore, no one was observed to spontaneously approach Musikiosk, despite attempts to introduce and demonstrate the system during Community Outreach. When it was in use; there was no visible reaction either to the music played nor to the Musikiosk users, except for ‘people-watching’, which had already been observed as a key activity of these park users before the installation (EO).

These regular users’ people-watching proved intimidating for some Musikiosk users (IM), one of which stated that:
“population-wise, you have regulars here who eye you every time you come in and you do feel like there’s regular people who are sitting here and they like their park the way it is, as in you coming in out of curiosity, wanting to try out this toy […] it does feel intimidating to come into this park”.(U2, F, 20s)


This sentiment was not shared by all Musikiosk users (IM): “I never noticed anyone giving me judge-y looks or any neighbors coming out with like dough-rolling pins being like “get out of here! You don’t belong here!” (U6, F, 36).

However, for the elderly Portuguese users, who usually sat talking amongst each other, this lack of reaction could be related to the position of the Musikiosk system in relation to their “usual spot” (EO): while the system was installed in the gazebo on the quiet, residential side of the park, their usual spot was on the commercial, active side of the park, where the music played through the system was frequently not audible, allowing for the creation of separate soundscapes dominated by different sounds. This was confirmed by Musikiosk users (IM), who stated that: “the sound doesn’t propagate to the surroundings, cause it’s not a very loud sound, but it’s just such an enjoyable part of being in a park” (U6, F, 36).

There were no measurable changes in the patterns of use of the park throughout Phase B (BMT), indicating that presence of Musikiosk did not lead to an observable transformation in the behavioural profile of the park (see [107] for a detailed analysis). However, observations (EO) did show that Musikiosk did increase the spontaneous use only of the gazebo on a daily basis, mostly by younger users, in small groups, engaging in lingering behaviours and using the system for hours at a time, sitting either in the gazebo or on its steps to maximize their listening experience (confirmed by IM). Also, the outreach events organized by the project team introduced new users to Musikiosk and increased the number of users performing socially interactive activities in the early and late evening in the park. Among the local users of Musikiosk, some of the participants who had come in the evenings returned, either as users of Musikiosk (alone or with a small or large group of people) or of the park itself (observed on different occasions e.g., reading on a bench or having lunch; EO).

One interviewee (IM) highlighted this dynamic between regular and new users, the way in which Musikiosk brings new people in without displacing the current users, while allowing for potential, but not mandatory interactions:
“I think they really didn’t mind [us listening to music through Musikiosk]. Because they were just continuing to do what they were doing, sitting down, playing with kids, talking. It didn’t really bother them. And I think that at the same time some of them were curious, like of what’s happening inside, because of some nice people having fun”.(U1, F, 27)


This diversification was largely perceived as a positive change, even by local community members, with one resident (IR) describing the process of how important it was to have new users for the viability of their community:
“and it also creates roots, the people who come, they know the neighborhood, maybe they’ll shop, eat closer […] It probably gathered more young people, who were pleasant, who were happy, and so people came to see what it was”.(R1, F, 79, see Table A6–6)


Another resident explained that they didn’t think this mix of people would have happened otherwise: “it’s an opportunity to sit and take a break outside, to meet people that we wouldn’t have met otherwise, indeed. No, it’s nice and it adds to the neighborhood life” (R3, F, 49, see Table A6–7).

In summary, Musikiosk was observed to add a new layer of activity and users into the existing park without displacing existing users. An explanation, in part, is in the way Musikiosk gave existing users something to people-watch. Musikiosk also occupied, sonically and physically, a small, but unused section of the park.

##### Interaction with Others


**Increased Interaction and Sharing**


In addition to lingering and the diversification of users and activities, the presence of Musikiosk encouraged interaction between people and sharing of information, music, and space.

Nine Musikiosk users, when reporting something they liked about the system mentioned how they liked the improved interactions with others, while nine others reported enjoying the feeling of sharing a space and/or music (Q2). Even among non-users describing the system, 3 spontaneously mentioned how it was good for bringing people together, and 3 others mentioned how it was good for sharing (Q3), e.g., “when someone came, they could share what they were listening to” (R1, F, 79, see Table A6–8).

One of the residents had used the system and talked about dancing with her children (IR): “it was merry, we were dancing […] And, the weather was nice, it was the end of summer, so it was festive” \ The same resident later explained, “I think it facilitated communication between us and the young people who were there before we arrived” (R3, F, 49, see Table A6–9 and 10).

A different resident (IR) explained in more detail:
“It’s great to have interactive devices in the city like that that can create community spaces that are a bit broader compared to simply meeting up. I imagine we can carry a wider ambiance with such a system […] It’s more participatory obviously, because the majority of activities on the Saint Laurent [boulevard] are very passive”.(R5, M, 34, see Table A6–11)


The aspects of increased interaction were visible when interviewees were asked about their Musikiosk experience or when they were asked about hypothetical situations in which others played music that they liked or not. They were generally open to actual or potential socialization with strangers brought upon by the sharing of music (IM): “If I play music out loud and someone else comes to me and talks to me, I guess I would talk to the person, that would be part of the whole experience, to meet people” (U3, M, 30). Two users described their encounters with other park users, curious about the system:
“A few people that were walking by came up and asked what the track was. One person wrote it down, and then two people that were sitting on a nearby bench, came up and were like ‘what is this? How do we make it work?’, and it was like ‘you just plug it in and play your music and that’s it apparently’, and they were like ‘cool, sweet.’ And then that was—we were probably here for fifteen minutes maybe, so there wasn’t tons of attraction but interactions with three–four people […] and they were enthusiastic about it”;(U5, M, 28)
“I just ended up getting involved in a conversation with two people I’d met before at Musikiosk, and then one or two people I hadn’t met and hung out for a while and then left”.(U4, M, 25)


One interviewee, acknowledging their reticence of usually interacting with others in public settings, welcomed the possibility of using music as a topic of conversation to interact with strangers:
“(if people came in the gazebo) I would probably ask them what they thought of the music, or maybe not say that right away but, you know, I’m not really one to start talking to people in public, but I’m thinking if I were the one playing music in here, I would feel more open talking to people who came by and sat down, you know. It’s obvious that they’re interested in the same things as me […] It changes the dynamic of approaching people in public a little bit in that case […] hopefully, that would generate some kind of conversation”.(U11, M, 24)


Interacting with others is described as an essential part of the music-listening process; as music is brought into the public sphere, the aspect of *sharing* the listening experience with others takes center stage:
“it’s more the fact that it’s a social experience of sharing the music that actually makes me enjoy being at the park. I’m not sure that if I just came here to read my book and I turn on the music, that I will actually enjoy the experience more. It’s really a matter of this being a social experience of sharing music”.(U2, F, 20s)


One interviewee explains this point in relation to her Musikiosk experience as shared with her coworkers:
“It was a really fun moment between coworkers, because everyone just kinda came out with a song on an mp3 player, we’d just listen to it, and we’d kind of see the person who’d played it and they’d get a little embarrassed, and then everyone’s laughing and listening to the song and discovering new music, which is super cool”.(U6, F, 36)


The nature of the Musikiosk system was evaluated by park users and residents alike as intrinsically encouraging social interaction and sharing, both of the space itself and of music. Actual or potential interactions, mediated my music and listening in a social, public context, were appreciated or welcome across the board. Two particular points were emphasized: self-policing—of music played (in relation to the sharing aspect) and the idea of “passing the baton” and sharing the actual use of the system.


**Increased Interaction and Sharing—Self-Policing**


As part of the interaction and sharing theme, the idea of Musikiosk users’ self-regulation of their musical behavior and choices emerged as a key finding. This was related to topics of “responsible listening” that have been addressed extensively elsewhere [113]. Musikiosk users (IM) intuited or “enforced” these rules on their own, relating them to being mindful of other park users and adjusting their content to an imagined “average other”, as a member of a musical community that shares the same ideas on what music would be “universally appreciated” (U2, F, 20s) or “nice for people to hear” (U7, F, 18):
“It turns off on its own so between those times, go for it, have fun, do whatever, as long as they understand there’s rules there, don’t be mean and loud and obnoxious cause there are people who live around it, but generally, live and let live, right?”;(U9, M, 26)
“Well I wanted to play something that would show how awesome my music tastes were, but also be something that a lot of people would enjoy and dance to”.(U4, M, 25)


Further demonstrating this idea of responsibility as part of sharing music and listening in public, interviewees were very sensitive to the possibility of someone *disliking* their music and quick in adjusting their content to the (perceived) preferences of the other listeners in the park, as well as to offer other the possibility to play their own music (IM):
“(if they didn’t like my music) I would ask if they have something else to play and they could play it after I’m playing my song or something”;(U3, M, 30)
“I mean, if it’s a music that is likely controversial, I would understand and I would turn it down. But if it’s just regular classical music, that 90% of the population seems to like Bach, Beethoven, Brahms, this type of—stuff like that. I won’t go with the hits because it really depends on the generation and I, myself, do not always agree with the charts but let’s say something very soft, non-aggressive classical music”.(U2, F, 20s)


Despite the open and democratic nature of the system that allowed its users to be free in their consumption of (musical) content, Musikiosk users self-policed and engaged in responsible listening practices as they emphasized the need to be mindful of other park users, their needs and tastes.


**Increased Interaction and Sharing—Passing the Baton**


In addition to self-policing, a sub-theme of interaction and sharing was the idea that Musikiosk users took on the responsibility of showing others how to use it. One of the residents (IR) who had used the system explained an interaction with a group that she was familiar with, but would have never interacted with:
“It had the potential of creating a nice ambiance, that could possibly gather the people who were a little in their individual world, it could unite maybe, because we started talking, actually with the young people who were drinking in their own corner, at any rate, it’s not people we would have initially approached, but then we explained to them what to do because they were listening to their own music but didn’t realize they could use [Musikiosk]”.(R3, F, 49, see Table A6–12)


*Passing the baton* was emphasized by an interviewee (IM), who acknowledged that for some park users the idea of engaging with the system spontaneously would be unlikely without having someone introducing them to the experience:
”It’s always fun to introduce people and it’s an easier way to get people to come and actually experience it, if they can latch on to something that they’re like ‘okay, I can invite people, and there’s gonna be a certain time, and there’s gonna be people and we can chill and hang out and understand what’s going on’, because if it’s just a thing, people might not touch it as much, if they don’t understand there’s something going on”.(U9, M, 26)


However, as mentioned above in the theme on interaction, interaction with others and invitations to system use were welcomed by most users as being a natural consequence of the system: “generally people would talk to you (when you are using Musikiosk) and then after a little bit, they’d ask to put something on.” (U9, M, 26). One interviewee described her experience:
“It wasn’t just me and my friends. It was actually also some strangers, who I didn’t know before and, not that I was planning to meet them, it’s just that they were there too and you can share, it’s not like it’s someone’s gazebo! Everybody can use it, so. I think one of the girls she played some Latin music so everybody just, all of a sudden, the mood just switched completely”.(U1, F, 27)


The fact that anyone could use Musikiosk to play their preferred content made its users conscientious about ideas of sharing and “passing the baton”; they both reported and were observed to encourage and show others how to engage with the system in order to allow everyone to appropriate the park sonically with their own content.

#### 4.2.3. Psychological Outcomes

The third and final dimension of QUPE on which Musikiosk had an effect was the category of psychological outcomes. The themes in this dimension pertained to individual effects on space users, which includes mood and restorativeness.

##### Mood

As was hinted in the previous section on passing the baton by a Musikiosk user, Musikiosk had a strong impact on moods. According to the questionnaires (Q1, Q2, Q3), users entering the park arrived with approximately the same mood across all contextual details (alone vs. others, age, etc.) and conditions (i.e., no significant effect was found on mood (before) for any variable.) Park visits increased reported moods significantly for all users (mood (after); see Steele et al. [109]), but the presence of Musikiosk boosted these moods even higher for both Musikiosk users and non-users (Q2, Q3). This effect was exaggerated for Musikiosk users (Q2), who saw yet significantly higher mood improvements.

Descriptions in the changes in mood from users (IM) varied from generic, “it put people in a good mood” (U8, M, 21), to the specific example above with the example of Latin music “switching the “mood”. The mood improvement was noticed even by a resident (IR), who said “It increased the turnout of people who were happy, but I can’t tell you the difference. People kept going there, families, children and all that” (R1, F, 79, see Table A6–13).

##### Public Space Evaluation—Post Musikiosk Lasting Effects

This part of the results section details findings from Phase C, when Musikiosk had been removed from the park. Memories and experience with the system and the park, as well as reflection on the engagement process lead us to some findings we term “lasting effects” of Musikiosk.

Interviewees (IM) cite their engagement with Musikiosk (both as system users and non-users) as a reason for a shift in their perception of Parc du Portugal:
“now I know this is a place that is friendly to me at least and I know, I’ve been here, I’ve used it, I know how to behave here so now for me. It’s a place where I can totally feel safe, I can come whenever I want, just sit down”;(U1, F, 27)
“now [...] I have associations […] that my friends would occasionally be (here), as opposed to, usually, when I would use this park before, I’d be by myself and I wouldn’t really be talking to anybody”.(U4, M, 25)


Thus, not only did Musikiosk increase the aforementioned lingering, but it also brought people to the park for the first time and created a lasting sense of belonging and safety.

Residents also described a positive change in their quality of life (IR):
“The more we inhabit public spaces, the more the quality of life improves. I think neighborhoods need to be animated, otherwise it creates dead zones […] It seems to me more pleasant to pass through a park with a few people playing music than nothing at all […] I think that yes, it broadened the hours of park use, so in that way, yes. It’s more pleasant and more convivial, a park with people compared to a dead park”.(R5, M, 34, see Table A6–23)


Conversations that took place with researchers (EO) confirmed that the space served well as a unique and lasting memory of tourists’ trips to Montreal, shaping their holistic impressions of the city.

Residents (IR) explained that Musikiosk brought a sense of safety:
“It’s a good use because it prevents other activities more […] I would even say criminal, from happening”;(R1, F, 79, see Table A6–24)
“There are a lot of areas like that, where it’s a little scary to pass through at night. If there was a community playing music regularly, at any time of the day, it makes it more pleasant and safer for everyone”.(R5, M, 34, see Table A6–25)


Resonating with the idea of safety, appropriateness for children came up often: “It’s a place for children, it’s safe” (R1, F, 79, see Table A6–26).

The lasting effects of Musikiosk thus revolved around a place and opportunity for making memories, and long-term impacts on feelings of belonging and safety.

##### Restorativeness

The final theme in the psychological outcomes dimension is soundscape restorativeness, described in the Review. Restoration contains five main elements, three of which applied to the study at-hand: fascination from sound sources, the soundscape offering a break from users’ routines, and the perception that “it’s easy to do what I want” in this soundscape. Note that Musikiosk questionnaire participants (Q2) were not directly asked to rate restorativeness.

Linked to the appropriateness of the soundscape for activity, Musikiosk had no effect on the perception of the park soundscape as being *easy to do what I want* (Q1, Q3), nor did Musikiosk have a significant effect across any other restorativeness scale; however, park restorativeness was already rated fairly highly, and it remained high with Musikiosk in place, demonstrating that the added sound sources did not compromise this aspect of the park’s character.

Free response and verbal data tell a more detailed story. Musikiosk users (Q2) reported appreciating something new (5), dancing (4), improved relaxation (3), and being invited to think (2). In a separate section, 4 users remarked on the system’s originality. Interviews (IM) elaborated many deeper, spontaneous mentions linking the system to increased restorativeness, particularly through a soundscape that makes it easy to do desired activities: “I can totally feel safe, I can come whenever I want, just sit down […], have a coffee with a friend, chill, maybe read a book” (U1, F, 27). A resident commented, “It’s out of the ordinary, it gives us an occasion to sit down and take a break outside, meet some people, some people we might not necessarily have met otherwise. No, it’s nice and it adds to the life of the neighborhood” (R3, F, 49, see Table A6–14).

### 4.3. Evaluation of Musikiosk

We turn now from examining QUPE as affected by Musikiosk, and instead focus on the interactions and experiences with Musikiosk itself.

#### 4.3.1. Engagement with Musikiosk

Musikiosk users (Q2) generally found Musikiosk easy to use and access, original, and found that it promoted activity. The simplicity and accessibility of the technology itself was appreciated by users:
“It’s nice when you don’t depend on anybody and you can just go there and enjoy it”;(U1, F, 27)
“So it’s a little music system, a very, very humble music system, basically five zero, plugged on the roof of a gazebo in a little park somewhere in Montreal, where you can play your own music and enjoy your own music, in an amplifier”.(U2, F, 20s)


User engagement has been discussed previously [113], employing O’Brien and Toms’ [115] process model of engagement to understand users’ approaches to music consumption through their engagement with Musikiosk as a free, democratic soundscape system. We showed how Musikiosk ultimately encouraged a feeling of temporary ownership of the space as well as a certain degree of complicity with the other users of the space, through the sharing of music as a social experience. Even outside of user engagement, one of the older residents (IR) described how the system made her want to engage more with technology:
“I want to come but I have a cellphone for emergencies […] I don’t know how, I’m not really equipped, I want to equip myself better, I would like to download all the music I like, and to be able to listen to it instead of on my little device, which is still cumbersome”.(R1, F, 79, see Table A6–15)


#### 4.3.2. Unrealized Fear of a Democratic Soundscape Intervention

One of the key challenges of Musikiosk was negotiating fears of the risks that residents, city authorities, and park users imagined would materialize as a result of the system. These fears were prominent particularly in the pre-project planning phase, as people imagined uses of Musikiosk that were contrary to their needs of the park. The borough authority was initially apprehensive of awarding the permit to install Musikiosk and required many rounds of negotiation and rule-setting, taking months (Municipal support and approvals). However, this process established a relationship of trust that was further solidified by the positive results of the system.

Elaborating on this unrealized fear from imagined system uses was a media outlet who had an interesting interpretation of what they believed the device would entail (Media Outreach): they photoshopped very large speakers facing outwards in every corner of the gazebo and claimed that “Montreal is getting a (massive sound system you can control with your phone” (https://www.mtlblog.com/news/montreal-park-is-getting-a-massive-sound-system-you-can-control-with-your-phone). After an exchange with the Musikiosk team, the news outlet removed the word “massive”.

Despite and perhaps thanks to the planning and coordination between the municipality and other stakeholders, in addition to the community outreach, very few problems came to pass. No formal complaints were lodged with the municipal complaint hotline (311). Residents (IR) spoke of this unrealized fear:
“It’s a concern, that once people are done listening to music, they will settle in the gazebo together to talk, some might bring beers, and it can maybe end later. Which can be invasive, but also very welcoming. That’s what we don’t know. Our experience at this time is that we didn’t have that, let’s be honest, it didn’t happen”.(R1, F, 79, see Table A6–16)


Unrealized fears were also seen from non-users (Q3), but attitudes toward Musikiosk varied greatly with degree of exposure to the system. When Musikiosk was not in use, the idea of the system gave park users pause; of 67 non-users, there were 9 negative judgments and all of them came at a time when Musikiosk was not in-use (condition: Musikiosk OFF). One fear was “I’m afraid that there might be too many types of music at the same time” (see Table A6–17). Non-users who were exposed to the system (Musikiosk ON), however, reported satisfaction, generally found the system appropriate for the park, and appreciated the unique offering of music in the public space; for example, the mood benefits of the Musikiosk ON subgroup were almost as high as they were for users (Q2), and were higher than the control condition (Q1). Non-users (Q3) who responded during Musikiosk OFF showed mood benefits that were not significantly different from the pre-installation condition (Q1).

If Musikiosk was in use, non-users (Q3) had less abstract concerns, e.g., “great idea, but I think the use of ‘public space’ for my personal enjoyment gives me second thoughts about using it”; “I don’t know if it’s necessary” (see Table A6–18); “It’s tricky because not everyone has the same musical taste”; and “good—as long as people are mindful of those nearby.” The concerns tie in with ideas of self-policing and responsible listening referred to above:
“If I was using the Musikiosk, it would be sort of with the understanding that it is a public place, and it’s sort of like a public place that I’m using, so it sort of feel like I should include people who wanted to come hang out. As opposed to sort of privately appropriate it for my own ends.”.(U4, M, 25)


In summary, we believe the “intuited” rules of Musikiosk involved the necessary negotiation of public space use such as to minimize the behaviors that were most feared.

#### 4.3.3. Appropriateness of the System for the Park

In addition to the Musikiosk creating a soundscape that made new activities possible, there was also a research question on whether Musikiosk was appropriate for Parc du Portugal.

40 of 41 Musikiosk users (Q2) found it appropriate for the park. Reasons for its perceived appropriateness were that it is well located, not too loud or disturbing, that it was friendly, and most interestingly, that it added something to the park that was missing. The one user who did not say yes said “yes and no”, and suggested that while it “bridges people together, some people like to meditate”—this reasoning did not appear to be associated with the park in question. One of the users who found it appropriate added that it was appropriate in the Parc du Portugal, but “not in my park”.

Users also weighed in on appropriateness during their interviews (IM):
“I guess that before it’s a really convivial park and it’s really nice, so I guess music is just a natural extension of the enjoyability of the park”;(U3, M, 30)
“I think the area is very dynamic, you have a very wide range of people that come by and also there’s a lot of noise coming from the city, you’re on a commercial street, there’s traffic, so it kind of gives a little serenity and it gives a little—it attributes almost like a, how would I say it, like to own part of the park or to feel part of it almost”.(U6, F, 36)


There were two interviewees uncertain of the appropriateness of Musikiosk for the park precisely because of the risk of displacing or disrupting “regular users” referred to above:
“[…] it’s sort of an older neighborhood that I don’t know if it’s really the sort of thing that a lot of the people who regularly use the park would do. I feel like there are other small parks where there are fewer old Portuguese people talking to each other in Portuguese on benches”;(U4, M, 25)


One resident (IR) was not pleased with the presence of the Musikiosk, principally because the park had been the host of a number of other festivals in the preceding months: “I was annoyed because there wasn’t only Musikiosk. All throughout the summer, we were bothered by the noise.” (R4, F, 33, see Table A6–19). Yet, to contrast this, a different resident explained his reason for liking Musikiosk was actually that having it there protected them from these more disturbing events, alluded to by R4, that happened at other times during the season: “It would seem reasonable, this thing that the city did a few times, […] to pretend to sing (The municipality had previously hosted a public karaoke event with a large speaker system.) …Yes, well that was noisy.” (R2, M, 95, see Table A6–20).

In line with the fair amount of agreement of the appropriateness of Musikiosk for this park, the above section on unrealized fears is also better contextualized. We again saw that those who had not interacted with Musikiosk were more likely to not find it appropriate. Also, the characterization of the park as previously “noisy” (as described in Section Soundscape Appropriateness for Activity) before the installation likely helped to prevent Musikiosk from disturbing a quiet environment.

#### 4.3.4. Uniqueness of the System

The uniqueness of the system was a recurrent topic in interviewee’s evaluation of Musikiosk, particularly in the context of its open, democratic, and political nature. “I think it’s a great idea, it’s the first time I’ve ever seen or heard of something like this, so I think it should become a permanent thing, for sure” (U7, F, 18); one resident summarized it: “It’s something out of the ordinary” (R3, F, 49, see Table A6–21).

The open, egalitarian and participatory nature of the system was emphasized by a number of interviewees. A resident remarked on what aspect was so different: “It’s the participatory aspect that’s different” (R5, M, 34, see Table A6–22). A Musikiosk user argued that:
“It’s free, it’s democratic, it’s ‘do whatever you want with that music.’ You do have a time limit, 30 minutes, just so that everybody has equal access to the music. It seems like a lot of fun and the people that thought of it are geniuses”.(U2, F, 20s)


A second Musikiosk user further focused on the political nature of the system in an urban context i.e., its democratic nature: “I feel like the whole nature, the whole point of Musikiosk is that it should be thought of as a public good […] it’s sort of a volunteering kind of public—everybody comes together to hang out. I feel that’s part of civic-mindedness” (U4, M, 25).

A third user highlighted the aforementioned idea of responsible listening, but showing how the system flips the script and offers park users control and trust to appropriate their space acoustically:
“I feel like it’s such a strong message, giving power to those who use this park, because in a sense, you give a tool—there are limitations to that tool, but you tell people ‘use it’ and ‘enjoy it’ […]—I think it makes it much more dynamic between the different types of people that use it. I’m really for that idea of good incentives, like being positive instead of negative and saying, ‘you can’t do this, you can’t do that’, just giving a tool to someone and being like ‘do what you want with it, we trust you’, it’s nice”.(U6, F, 36)


Musikiosk was also perceived as offering park users a unique way of interacting with the park, to do something that’s normally not permitted, namely, play music out loud in a public space: “there are speakers available to anyone and a really nice park, and anyone can join and put their music on if they like and enjoy the park as they please, but with […] added music of their liking. It’s not like a program…, you go to listen to your music in a park, officially, sanctioned by the city” (U3, M, 30).

## 5. Discussion

The goal of this paper was to investigate the effects of a democratic soundscape installation on various aspects of public space use and appropriation in an urban pocket park, utilizing the proposed concept of QUPE and its three axes. Considering that the nature of Musikiosk directly affected users’ soundscapes in their public space, we started by focusing on the sound-related aspects of evaluation of one’s experience, but then demonstrated that the effects of Musikiosk were more far reaching, as we studied broader changes in users’ engagement with the park as well as potential outcomes at a psychological level.

We first discuss the effects of Musikiosk on each of the three-subthemes of QUPE, then reflect on methodological and theoretical contributions that this paper (and Musikiosk) brings.

### 5.1. Soundscape Evaluation

The appropriateness of the soundscape for the reported activity of users was in any case rated as high across conditions and remained unchanged (high) by Musikiosk; in fact, appropriateness of the soundscape was the highest rated of all descriptors (including pleasantness), further demonstrating the added value of a pocket park, even if with one side on a comparatively loud, busy street, for the urban public experience. Extending research that shows that the level of social interaction of one’s activities influences their evaluation of soundscape [31,40], Musikiosk users stated that Musikiosk was best used with others, encouraging social interaction. Furthermore, in line with the questionnaire findings, interviews showed that the system was perceived to encourage interaction with strangers, as well as solitary enjoyment of the park, maintaining high appropriateness of the park for a range of activities.

The present study extends findings from Steffens et al. [50], which showed benefits in pleasantness and eventfulness when people were modifying their own sound environments with their own music. The difference is that these participants were usually modifying home or office environments with music, but Musikiosk brought these benefits to a public park

In terms of the effects of specific sounds, Musikiosk encouraged soundscape benefits in terms of, for example, masking of traffic [25,41,116]. Unlike other soundscape installations that achieved that with purposeful acoustic design [25,37,41,42,43,44], Musikiosk did not have pre-programmed content. However, users displayed care in the content they chose to play (generally, music), engaging in self-policing behaviors [113] in relation to the genres of music they played and what was considered appropriate for the setting of Parc du Portugal and its users.

### 5.2. Engagement with the Park

Our mixed-methods study confirmed the well-documented aspect of the sociability of public spaces, acting as contexts for different forms of social interaction [54,58,75]. Interviews with local residents, park users in general and Musikiosk users in particular, as well as behavioral mapping and ethnographic observations showed that Parc du Portugal was used (and evaluated as suitable) for socially interactive activities, but also for restoration-related activities, in line with the findings of Peschardt and Stigsdotter [93] on urban pocket parks. There were clear temporal patterns in how the park was used, that influenced the profile of users; there were elderly regulars that had been identified in the preparation stages of this study and recurrent users of various ages observed during lunchtime (usually middle aged) or in the evenings (usually younger). Considering that the soundscape in the park was evaluated as appropriate across users’ diverse activities for all time periods, we employ Peschardt’s findings [93,94] on the provision of park amenities. We show how the provision of an amenity that allowed for public music playing that encouraged socializing benefited park users, ensuring that restoration was also possible [95].

Musikiosk resulted in increased and diversified social interactions in the park without disrupting ongoing activities. It also brought new users while not displacing the regular ones. Musikiosk offered users control over the content played, which led to the creation of temporary public auditory communities based on shared listening practices, along the lines of the work of O’Hara and Brown [117]. These temporary communities were described to function under self-imposed rules of responsible listening, as described by Haake [104], as well as the need to share the access to Musikiosk and teach others how to use it by “passing the baton”.

The engagement with the Musikiosk system itself, as mediated by mobile music listening technologies like smartphones [65,118] further encouraged aspects of interaction by e.g., offering the opportunity to exchange and share music, discuss musical tastes and opinions as well as listening together. The democratic and free nature of the system allowed them to bring their private (musical) identity to the foreground and into the public realm, along the lines of Thompson’s [69] suggestion on the role of mobile music technologies in blurring the line between the private and the public realm.

### 5.3. Psychological Outcomes

The benefits of a park visit on psychological outcomes have been demonstrated in the literature [57,75,76,77,78,79]; the sounds of parks have also demonstrated a restorative effect [82,83]. In addition, the aforementioned Steffens et al. [50] study showed positive benefits of the presence of music on mood. In the present study, we saw a positively interacting effect on mood and restorativeness of both the park visit and the presence of one’s own music, which may not have been predicted using a sound level-based approach [71,72].

The engagement with Musikiosk was reported to have affected park users’ emotional engagement with Parc du Portugal [74], triggering (positive) long-term effects on their evaluation of the park, particularly in terms of psychological comfort [57] and perceptions of safety and appropriate behaviors for the space, shifting previous negative evaluations of the park. Some Musikiosk users specifically addressed issues of control over their public space, in relation to more general ideas of access to and expected behavior in public spaces in general (along the lines of the work of Atkinson [79]). Also, in relation to the (unique) opportunity to play and the (unique) opportunity to have control over their auditory spaces [102], Musikiosk users managed their soundscapes by playing their own amplified content in a public space context.

### 5.4. Contributions

On methodological grounds, we employed a mixed-method approach to better capture and understand the impact of Musikiosk as well as to situate the results and implications of different methods. Further, we needed a mixed-method design to be able to properly capture the three axes of QUPE. These methods came from a number of disciplines, as was necessary to adequately capture aspects of e.g., soundscape evaluation as well as the social, behavioral and psychological outcomes resulting from park users’ interaction with Musikiosk. For example, while behavioral mapping did not show a substantial change in the patterns of use of the park, ethnographic observations showed a subtle yet clear diversification in terms of users and an increase in lingering times, particularly in the area around Musikiosk, driven by the engagement with the system. Interviews allowed us to go in depth in some of the topics that were touched upon in the questionnaires, to further investigate aspects of engagement with the park or evaluation of the system that could not be captured using quantitative data collection methods.

On theoretical grounds, we created the QUPE model to integrate the auditory and non-auditory aspects of a soundscape intervention and to make a connection to urban public experience and public health. Our findings converge to show that a democratic soundscape installation affected not just the soundscape in the park but broader aspects of use and engagement with the park as well as outcomes at a psychological level, increasing the perceived pleasantness and restorativeness of the park and intensifying as well as diversifying interaction in the park without displacing or disrupting existing users or patterns of use. From another angle, we also contribute to a growing body of work on the restorative, social, and cultural functions of small urban spaces, i.e., pocket parks, as distinct from the more studied large, green parks.

On practical grounds, the benefits to QUPE were achieved through a relatively modest, low-cost intervention. For future urban planning and design efforts, such interventions could extend the range of solutions available to address tensions between commercial, residential, and leisure uses in dense urban areas.

## 6. Conclusions

Our study provides insights into understanding how a democratic soundscape installation can affect the quality of the urban public experience of space users, and how goals of e.g., restoration and increased social interaction can be attained and accommodated in the context of a small, urban pocket park. The findings presented here highlight the integration between multiple data collection methods as they contribute, in a complex way, to the stated research question. Findings from individual methods have and will be presented in separate studies (see [22,108,110,114]). Future work will also include an analysis of acoustic measurements taken throughout the study. In preliminary analyses (e.g., descriptive statistics, mediation analysis), the acoustic measurements (referred to in Figure 2 as EMM: Environmental Monitoring with Microphones) did not prove predictive or descriptive of other themes, such as the soundscape evaluation scales and restorativeness. Future research will integrate the analyses of the EMM using, for example, manual annotation of recording data or computational auditory scene analysis. Future studies based on the Musikiosk model will attempt to address the constraint that the microphones were placed very close to the sound installation.

This paper illustrates a number of contributions put forward by Musikiosk as a democratic soundscape installation to both sound-related scientific knowledge as well as to city-making knowledge, particularly for designers and other city workers working on small-scale urban projects concerned with the quality of the urban public experience of city dwellers in their encounters with and use of public spaces. From a methodological perspective, Musikiosk was a proof of concept on the process of developing both a complex installation for use in an urban pocket park context and an integrated mixed-methods research strategy to document the effects (if any) of Musikiosk on park users’ quality of their urban public experience. Considering that Musikiosk is an intervention that directly influenced the park soundscape, it was necessary to investigate the sound-related effects it had over users and their evaluations; however, its effects were farther reaching. QUPE, having been developed iteratively, could thus be a framework that stands for the missing link from quality of life to public experience and it could be extended to other aspects of the multisensory urban public experience; we focused on sound, but it could very well be applied to other aspects of it.

Partially going against mainstream policy-oriented quests for noise abatement where urban sound is conceptualized as “bad for public health”, we confirmed the findings of previous initiatives centered on adding decibels through soundscape interventions in public spaces. However, unlike most known such initiatives, that are based on top-down processes of auditory output selection (in which usually sound artists, alone or with the support of local stakeholders, design the content to be played), Musikiosk was developed with ideas of democratizing the (auditory) appropriation of public spaces by allowing the public space—and system—users to choose their own content, musical or not. From a technological perspective, the aspect of interacting with the installation itself was not essential; the technology behind Musikiosk was developed to be simple and user-friendly, so as for it to be a means for city dwellers to engage with others and with the park itself through the aforementioned appropriation of the public space rather than an end in itself.

The democratic aspect of the installation represented the main source of both interest and apprehension for its users, and that brought with it complex negotiations with one’s self in terms of “adequate” music to play for the other park users and the park itself while adhering to principles of mutual respect. We can thus argue that the availability of such a democratic sound system that allowed users to control (to an extent) their soundscape, contrary to expectations, tapped into an intuitive, perhaps empathetic understanding that sharing their music openly would make their private domain (and, to an extent, their identity as defined by musical taste) public; thus, potentially because they have had their auditory spaces invaded previously (by music blasting from cars, motorcycles revving, construction work), Musikiosk users were conscientious in their engagement with the system.

Despite repeated worries expressed by officials as well as acknowledged by multiple members of the public (particularly Musikiosk users), the minimally invasive nature of the system, combined with its conscientious use was confirmed in the results of questionnaires as well as the observed patterns of use of the park, which did not change significantly when the system was in use per se, nor overall, in the post-intervention phase. What Musikiosk did succeed in was diversifying the park i.e., bringing new users to the park by changing their previous impression of the park (as a consequence of their engagement with the installation), while not displacing the current users, particularly members of the local aging Portuguese community, who did not show signs of interest in engaging with it throughout the intervention stage.

Finally, the effects of Musikiosk on QUPE have been observed and recorded in a small spatial context i.e., Parc du Portugal, a pocket park in a residential area off a busy traffic artery in Montreal; while it is well known and accepted that access to and use of large urban parks have beneficial effects over the health and wellbeing of urbanites, this project demonstrated that similar benefits in terms of restorativeness, mood or diversification of users can be achieved from a small (pocket) park. Thus, key aspects in designing a restorative environment are that it has to match the activities envisaged, users should be granted more control over their environment, work closely with the community to make sure that they will be on board with plans. Furthermore, our findings indicate that the effects of the installation extend beyond only those who interact with the system. This insight is of particular interest for urban designers and other professionals intervening in city design because it demonstrates that increasing the quality of the urban public experience of city dwellers is an attainable goal at low costs by developing and providing sufficient access to small, yet plentiful and smartly distributed spaces of respite and enjoyment, exposed to the sounds of the city. In these contexts, given the spaces themselves and the types of activities either current or intended, installations like Musikiosk could be low cost yet democratic alternatives to more complex soundscape installations that can provide public space users with the responsibility of using and appropriating their spaces. The findings potentially suggest that this type of intervention may be appropriate for similar public spaces, i.e., those with a relatively noisy character and dense activity, where sound levels are elevated but not harmful, and there is no “quiet” to disturb. This approach is not intended as a replacement of noise mitigation measures, but rather could serve as a complement by addressing aspects of quality of public life and accounting more deeply for users’ perceptions.

## Figures and Tables

**Figure 1 ijerph-16-01865-f001:**
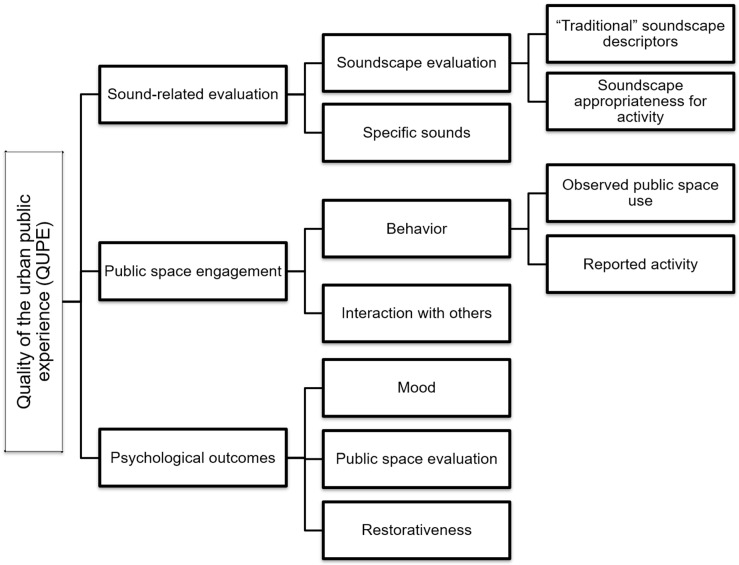
The QUPE (quality of the urban public experience) model. Three axes are elaborated and inform the structure of the present study.

**Figure 2 ijerph-16-01865-f002:**
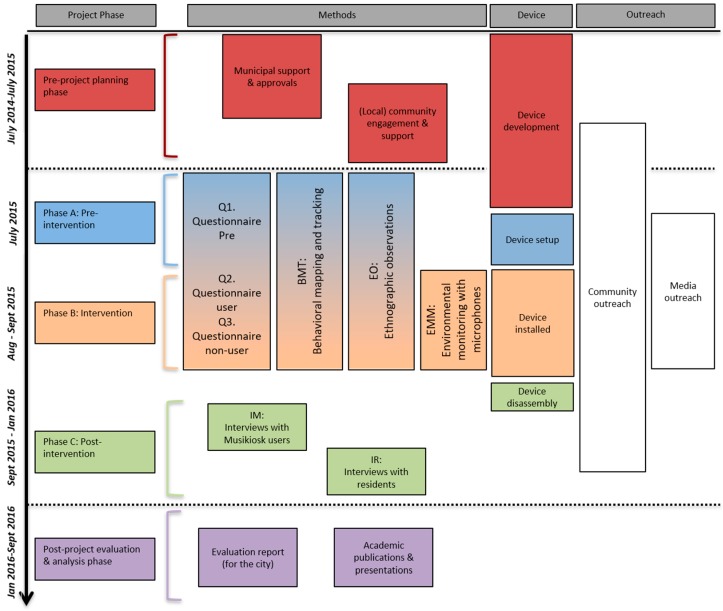
Musikiosk research timeline by phase. Legend indicates method type.

**Figure 3 ijerph-16-01865-f003:**
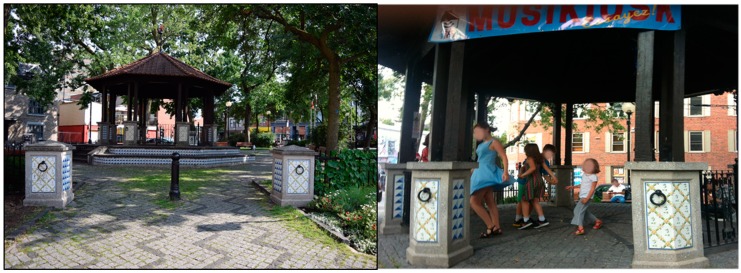
Left: gazebo in Parc du Portugal; right: users enjoying Musikiosk.

**Table 1 ijerph-16-01865-t001:** Research methods supporting QUPE analysis. Secondary methods support the results of the primary methods in this mixed-method study.

QUPE Theme	Sub-Theme	Supporting Methods
Soundscape evaluation	‘Traditional’ soundscape descriptors	Primary: Q1, Q2, Q3 Secondary: IM
	Soundscape appropriateness for activity	Primary: Q1, Q2, Q3 Secondary: IM, IR
Specific sounds		Primary: Q1, Q2, Q3 Secondary: IM
Behavior	Observed public space use	Primary: Q1, Q2, Q3 Secondary: IM
	Reported activity	Primary: BMT, EO Secondary: IR, EO, Q1, Q3
Interaction with others		Primary: IM, IR
Mood		Primary: Q1, Q2, Q3 Secondary: IM, IR
Public space evaluation		Primary: Q1, Q2, Q3 Secondary: IM
Restorativeness		Primary: Q1, Q2, Q3 Secondary: IM

**Table 2 ijerph-16-01865-t002:** Activity categorization before and after Musikiosk installation. Note: if people reported multiple activities, it was coded more than once.

Activity	N (Q1)	N (Q3)
Consuming (eating, drinking, smoking)	42	43
Relaxing (resting, waiting, sitting, sunbathing)	39	28
Interacting (date, meeting friends)	31	35
Moving (walking around, biking)	12	13
Observing (people-watching)	4	4
Music (Musikiosk, dancing, listening)	0	14

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
