# Peer review of "Soundtracking the Public Space: Outcomes of the Musikiosk Soundscape Intervention"

_ijerph, 2019, doi:10.3390/ijerph16101865_

Round 1

Reviewer 1 Report

     I would like to start the review with a quote from R. Murray Schafer, who many believe coined the term Soundscape, in his 1977 book "The Tuning of the World." "Ultimately, this book is about sounds that matter.  In order to reveal them it may be necessary to rage against those who don't."I would suggest that the authors of this paper read R. Murray Schafer's books and writings.  He did not refer to people who battled the noises as demonizing loudness.  In the article, one sentence is devoted to the deleterious impacts of noise on health and then generally dismissed by the following sentences which appear to criticize those people who are stressing the dangers of loudness and noise.  I have read many articles, academic and popular, on adverse noise impacts and the authors of these papers generally recognize the beauty of the good sounds and stress that these will be lost unless we lessen the intrusive loud and noisy sounds.  If indeed the authors wish to engage people in the practices covered by their study and enlist their involvement, I believe they should consider rewriting the introduction. 

     However, it is not simply the introduction that is problematic.  With the ideas as noted above that the authors bring to this study, some doubt might be cast on the on their methodology and their interpretation of the comments of the respondents.  The words used to describe older people who may seek some quiet in parks is also troublesome.Yes, music can be soothing and comforting but some people seek out parks to relax and enjoy the natural sounds surrounding them.  Also in today's world, one can bring one's own music to be listened to privately through earbuds. 

     I believe others will find this study worth reading and I know the authors have presented their findings at conferences but I again stress the tone needs to be changed, especially if these authors wish to have a wider group explore these practices.

Reviewer 2 Report

The authors have studied the effects of the spontaneous use of a Musikiosk on the soundscape of a pocket park. The study results sound but it still needs further elaboration and presentation of the data. The structure of the paper should be also revised considering that excessive sublevels of the paragraphs negatively affect its readability. Please highlight the boundaries within which this installation could be effective.

In detail:

- The paper is well written; however this reviewer finds it very long (please check the journal limits on this aspect); moreover, I strongly suggest to avoid reporting the text within brackets: it does not help to clarify at it makes the reading flow contorted.

There are different rather unuseful paragraphs titles that do not have any information. Please reconsider the structure.

- Figure 1: are these measurable? The surveys results do not give any quantitative data or are not presented even if some of the questions are in a likert scale. No data are shown from Appendix A questionnaires. How is considered/evaluated health?

-line 161: anticipation of the results should be avoided.

-line 169: serve or see?

- from 2.1 to 2.5: the paragraph should be coherent and give the relation to the Musikiosk case-study (e.g. as it is done in 2.3.3).

-2.3.2: the following paper is missing: https://www.igi-global.com/chapter/influence-of-soundscapes-on-perception-of-safety-and-social-presence-in-an-open-public-space/198159.

- line 198 vs line 202: which is the difference between restoration and restorativness?

-At the end, the authors should clarify how the results are related to the introduced concept of QUPE.

-How could the results of this study be used? It seems that the answers to the interviews suggest a very strong effect of the individual connection to this space.

-line 396: dot is missing after bracket

- 3.1: a foto or other graphical representation of the musikiosk and the park should be given

- Line 454: please mention here the software.

- Figure 3 is not sufficiently commented and it is not clear how it has been used. Are all the data presented in this way for all the monitoring?

-lines 1153-1154: these should emerge and be clarified better

-the number of interviews with the nearby residents seems very low

- please check and correct the order of the references within the text e.g line 144.

- all the submitted references and not yet accepted should not be reported

- line 726: now should be know

Round 2

Reviewer 2 Report

Thank you for revising the paper according to the previous suggestions. 

However, point 4 remains still unclear. Since it is not a trivial part of the paper, it should be clarified in the text of the mansucript (besides the answer to the reviewer).

Author Response

Point 4: Figure 1: are these measurable? The surveys results do not give any quantitative data or are not presented even if some of the questions are in a likert scale. No data are shown from Appendix A questionnaires. How is considered/evaluated health?

Response 4: Each of the QUPE themes and sub-themes is connected to observable entities, by design. They have been elaborated as axes and themes to help conceptually bridge the gaps between findings from each of the methods. We feel that the highly structured review and results make it clear for the reader to understand these links.

Point 4: part 2: However, point 4 remains still unclear. Since it is not a trivial part of the paper, it should be clarified in the text of the mansucript (besides the answer to the reviewer).

Response 4: part 2: In the previous response, the authors failed to point out the addition of Table 1, which shows the links between the QUPE themes and sub-themes with the methods that supported their inclusion in the model. Specific clarifications have been added throughout the document to clarify the sources.